# IPdb: A High-Precision IP Level Industry Categorization of Web Services

Paper #1896, 8 pages body, 12 pages total

## ABSTRACT

IP addresses with web services are crucial in the Internet ecosystem. Classifying these addresses by industry and organization offers valuable insights into the entities utilizing them, enabling more efficient network management and enhanced security. Previous work in website classification and Internet management struggles to offer an IP-level perspective of the industries of web services due to their limited industry categories or potential industry inconsistencies between IP address owners and AS owners. To this end, we present IPdb, an IP-level industry categorization dataset. To construct the dataset, we developed LLMIC, a Large Language Model-based Industry Categorization framework with a precision of nearly 96%. IPdb serves as a labeled database for future endeavors in developing IP-level industry classifiers, encompassing over 200 million IP addresses. Furthermore, our study indicates that 30% ∼ 50% of organizations within critical infrastructure industries deploy web servers across multiple ASes. Our study also validates the problem of mismatched granularity in industry categorization at the AS level with 87.83% ASes in IPv4 and 72.96% ASes in IPv6 containing IP addresses from different industries.

### ACM Reference Format:
Paper #1896, 12 pages. 2024. IPdb: A High-Precision IP Level Industry Categorization of Web Services. In *Proceedings of ACM Conference (Conference'17)*. ACM, New York, NY, USA, 12 pages. https://doi.org/10.1145/nnnnnnn.nnnnnnn

## 1 INTRODUCTION

IP addresses form a foundational element of cyberspace, with those hosting web services playing a particularly critical role. Identifying which organizations and industries manage these IP addresses sheds light on the roles they play within the broader Internet ecosystem and provides opportunities to analyze their traffic patterns [6, 11, 20, 23, 41]. Furthermore, web servers linked to critical infrastructure industries are often prime targets for cyberattacks [1, 24, 28, 35, 51], making the accurate identification of these servers essential for assessing and strengthening the security measures.

Previous studies have often concentrated on a limited set of specific organizations [6, 11, 20, 24, 41], a scope insufficient for conducting industry analysis across a broader range of web servers. Alternatively, some research has relied on Autonomous Systems (ASes), which represent collections of IP addresses managed by a single entity, to infer IP address ownership [23, 40, 46]. Research

focused on AS ownership [3, 5, 10, 15, 16, 57] has yielded valuable insights into the Internet ecosystem at the AS level by addressing issues such as BGP instability across industries. However, there are instances where organizations lease IP addresses from network providers to host their web servers. For example, as indicated by its disclosed IP range list [2], Amazon AWS hosts some of its web services on AS6167, owned by Verizon. Therefore, focusing solely on the AS owner is inadequate for answering critical questions about IP addresses, such as "Which industry has vulnerable web services on its IP addresses?" or "Which organizations deploy web services across multiple ASes to mitigate cross-AS traffic costs?" A comprehensive understanding of IP address ownership within AS boundaries is crucial for accurately addressing these questions. Nevertheless, the lack of annotated IP-level datasets remains a significant barrier, hindering researchers from conducting detailed analyses of the industry-specific distribution and security posture of web services at this granular level.

In this study, we introduce IPdb, an IP-level industry categorization dataset that includes both IPv4 and IPv6 addresses. The dataset is constructed using IP addresses as input, with reverse DNS queries employed to retrieve the associated domains. The organizations operating these domains are identified through Whois records and certificates. For industry classification, information extracted from the Web, specifically Wikipedia [54], is consulted to gather information on the primary business of each organization. Given the demonstrated proficiency of Large Language Models (LLMs) in handling complex Natural Language Processing tasks [8, 17, 39], we leverage these models to classify organizations by industry based on the Wikipedia descriptions. However, due to issues like hallucination, LLMs can sometimes produce uncertain and incorrect results. To address this, we propose a Large Language Model-based Industry Categorization framework (LLMIC) to improve the accuracy of categorizations. This framework integrates multiple LLMs, fine-tuning their combination weights on a manually labeled dataset to enhance performance through the complementarity of different models. Our system achieves outstanding results, with a precision of nearly 96% on the manually labeled *Gold Standard* dataset of organizational records. Using this system, we have expanded the dataset to include over 200 million IP addresses associated with web servers. This annotated dataset provides researchers with valuable data for web measurements analyzing the industry affiliations of IP addresses, aiding in Internet management and security efforts.

Our contribution can be summarized as follows:

- We developed an LLM-based industry categorization framework that achieves a precision rate of nearly 96% using a limited set of annotated data.
- We built IPdb, an IP-level industry classification dataset containing more than 200 million IP addresses, to provide comprehensive support for IP-level industry classification.[1]

---

[1]The dataset and code are available at github.com/IPLevelIndustryDB/IPdb.

- Our study indicates that 30% to 50% of organizations within critical infrastructure industries deploy web services across multiple ASes to optimize service delivery to end-users.
- Our analysis reveals a significant discrepancy between IP-level and AS-level industry categorization, with 87.83% of IPv4 ASes and 72.96% of IPv6 ASes hosting web servers from industries outside the AS owner's designated industry. This cross-industry phenomenon is widespread across ASes owned by organizations in various industries.

## 2 CATEGORIZATION STANDARD

Previous efforts [15, 16] about AS-level industry categorization offer a rudimentary system with a limited number of categories, such as Internet Service Providers and Universities. Critical infrastructure industries like Energy, are often omitted from these classifications. Despite the introduction of the more inclusive NAICSlite system by ASdb [57], based on the North American Industry Classification System [4], this system introduces redundancy by including non-critical industries. This issue is encountered in most commercial categorization standards [4, 32, 50], as they concentrate on assessing economy and commerce rather than network management.

Motivated by the need to capture critical industries for Internet management, we introduce our novel categorization standard, denoted as GICSmod, to mitigate the shortcomings of current classification standards. This system is a modification of the Global Industry Classification Standard (GICS) [32], focusing more on the importance of critical industries and decreasing the emphasis on non-critical industries as appropriate for Internet management. For example, in GICS, *Financials*, a level-1 category, is subdivided into 18 categories based on distinct production processes. While the level-1 category holds significance in Internet management, the further subdivision into subcategories holds lesser relevance, hence we maintain only the *Financials* category without further division. Conversely, as for the *Information Technology* category of GICS, encompassing *Software*, *Internet Service & Infrastructure*, and other sub-sectors with indispensable roles in Internet management, we retain these critical sub-industries and extend the *Cybersecurity* category within the GICSmod to emphasize their importance.

GICSmod is a streamlined, one-layer approach comprising 23 categories, significantly fewer than those offered in NAICSlite [57]. This redundancy reduction facilitates categorization while retaining essential categories for Internet management. Similar to previous categorization standards, the categories are defined to be mutually exclusive to avoid ambiguity, ensuring that a particular business practice does not fall into multiple categories. For instance, the *Service* category, following the GICS definition, includes only services not classified under other categories, such as human resources and legal services. Services that fall under other critical infrastructure industries, like IT and educational services, are excluded from the *Service* category. It is important to note that these definitions do not restrict an organization to a single category; for example, if an organization provides both legal and IT services, it should be classified under both the *Service* and *IT Services* categories. A detailed description of GICSmod is provided in Appendix B.

## 3 METHODOLOGY OF CATEGORIZATION

In this section, we outline our methodology for IP-level industry categorization. Since categorization essentially involves categorizing the organizations that own them, we begin by identifying the organizations managing web services on the IP addresses. These organizations are then classified into industries. The framework of our methodology is outlined in Figure 1. First, we use reverse DNS queries (i.e., DNS PTR records) to establish correspondences between IP addresses and their associated domains. Next, we identify the organizations that own these domains by performing domain WHOIS lookups and examining the subject fields in the domains' certificates. Then, we scrape descriptions about the owner organizations from Wikipedia [54]. Lastly, we introduce LLMIC, combining LLMs to classify these organizations into their respective industries.

### 3.1 IP to Domain

Reverse DNS lookups are used to query DNS servers for domain names associated with specific IP addresses. This process relies on DNS PTR (Pointer) records, which are configured by the IP address owner and stored in the DNS reverse zone file. The reliability of PTR records in determining IP address ownership has been proved by previous studies [29, 38]. To avoid redundant Internet scanning, we utilize the comprehensive IPv4 reverse DNS data provided monthly by the ipsniper project [25]. For IPv6 addresses, however, due to the limited availability of reverse DNS data on Internet measurement platforms during our research, we actively performed lookups using our own probers. Given the vast IPv6 address space, scanning it entirely was impractical, so we focused on IP addresses from published IPv6 hitlists [19, 44, 45, 56] for our reverse DNS lookups.

Although the forward DNS lookup is also commonly used to determine the associations between domains and IP addresses, they are not suitable for our research due to the potential discordance between the domain owner and the IP address owner. In cases where organizations provide Content Delivery Networks (CDNs) or shared hosting services, this discordance becomes apparent. As illustrated in Figure 2, while the IP address should be attributed to the CDN vendor, a forward DNS lookup may incorrectly associate it with the domain owned by the user. Although these IP addresses provide web services for users, they are highly variable and not under the control of the domain owners. In this case, we consider these IP addresses as assets of the CDN vendors, since they are effectively controlled by the CDN vendors, who decide the domain under service. Conversely, DNS PTR records, configured by the IP owner, correctly point to the domain owned by the CDN vendor, thereby accurately reflecting this ownership.

### 3.2 Organization Identification

The domain retrieved from DNS PTR records is the fully qualified domain name (FQDN), which represents the complete domain name for a specific host. Since identifying the organization for each FQDN is challenging and labor-intensive, we extract the *main domain* from the FQDN. A main domain (e.g., example.co.uk) contains a suffix under which Internet users can directly register names, including top-level domains (e.g., .co) and country code second-level domains (e.g., .co.uk), prepended by the name registered by the user. With the assistance of the public suffix list [27, 31], we identify the suffix

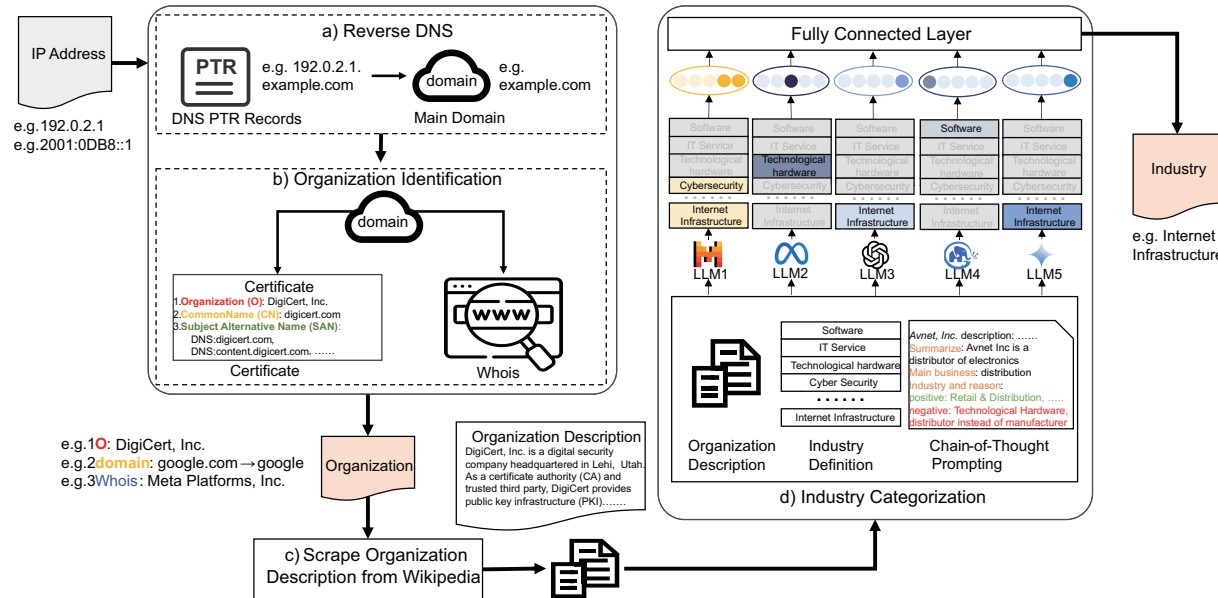

**Figure 1: Methodology for Categorizing IP Addresses into Industries**

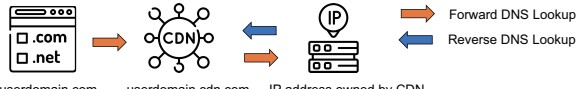

**Figure 2: The process of forward DNS lookup and reverse DNS lookup in a case where IP addresses provide CDN service**

and the main domain of FQDNs. We then identify the organization at the main domain level.

Since publicly available domain WHOIS data is maintained by domain registries recording the registrants' organizations, we use this data to determine the owner organization of the main domains.

Due to privacy concerns and the General Data Protection Regulation [49], some organization information in WHOIS data may be withheld or concealed. Consequently, SSL certificates are used to obtain the owner organization in cases where WHOIS data lacks registrant information. We query port 443 of the *www* subdomain under the main domain to obtain SSL certificates, which contain essential information about the domain's controlling organization. The *Organization* field in the subject indicates the organization holding the issued certificate. This information is validated in organization-validated and extended validation certificates, ensuring its reliability and trustworthiness. However, some domains hosted on CDNs may use certificates held by the CDN vendor, leading to a misalignment between the certificate owner (CDN vendor) and the domain owner (CDN user). To address this issue, we manually identify CDN and cloud vendors using descriptions from Wikipedia [54]. Given the large number of organizations, judgments from large language models (LLMs) are used as a reference to support manual work. If the Organization field in a certificate matches entries on a blacklist, it is ignored. Otherwise, the information is extracted as the domain's owner organization. Other fields, like the *Common Name* field and the *Subject Alternative Name* field, provide the domains authorized by the certificate. These fields can also provide information about

the owner organization. However, due to the use of shared certificates by CDN users, these fields may include domains belonging to the CDN vendor or other users, making the information they provide less reliable than the requested domain name itself. For certificates where the Organization field is vacant or ignored, the main domain is used as the identifier of the organization for further organization information retrieval.

### 3.3 Organization Information Retrieval

After identifying the organizations, we search for their descriptions on Wikipedia [54] via search engines. To ensure that the search results correspond accurately to the queried organization, we extract keywords from the organization's name, omitting common suffixes like *LLC*. Results that do not contain the keyword in their text are filtered out. Among the remaining results, we prioritize English Wikipedia entries. If no English Wikipedia entry is found, the top search result is selected and translated into English [22].

### 3.4 Industry Categorization

While some Wikipedia descriptions directly suggest the pertaining industry, most merely outline the principal business activity. The complexity of natural language processing tasks makes it infeasible to handle large amounts of data manually, necessitating the development of a categorization system to determine the industry.

**Challenges:** Firstly, the scarcity of available annotated data poses a significant barrier. Despite previous studies [5, 15, 16, 57] considering the industry of an AS, the resulting annotated data remains inadequate and imbalanced for IP-level categorization. Secondly, the inherent ambiguity of natural language and the inconsistencies in the writing styles further complicate the categorization process. Finally, the diverse principal business practices among organizations within the same industry introduce additional challenges. The scarcity of annotated data impedes training a deep

neural network from scratch, as they require a substantial quantity of annotated data. Conventional machine learning algorithms like Support Vector Machines and Random Forest also exhibit poor performance on this specific task, because of the ambiguity and semantic divergence of the input descriptive texts.

Large Language Models (LLMs) offer promising results in handling such complexity. The pretraining process on extensive general corpora and the supervised fine-tuning process with human annotations enable the acquisition of natural language processing capabilities. These models can then adapt to our specific task using only a small amount of annotated data for few-shot learning. However, LLMs can sometimes produce unsatisfactory outcomes due to issues such as hallucination, where the model generates incorrect or fabricated information.

To mitigate the problems, we propose an LLM-based Industry Categorization framework, referred to as LLMIC, which enhances the ability of individual models and leverages the capabilities of multiple LLMs to determine the industry of organizations. The LLMs integrated into the framework can be substituted to improve performance or meet specific requirements. Organization descriptions sourced from Wikipedia, along with industry definitions within GICSmod, are input into the integrated LLMs. For LLMs that support multiple languages, the original information from Wikipedia is provided regardless of its source language. For LLMs that only support a limited number of languages, including English, the English-translated descriptions are supplied.

Previous work [53] has demonstrated the effectiveness of the Chain-of-Thought prompting technique in enhancing the reasoning and problem-solving capabilities of LLMs. This technique involves breaking down a task into smaller, sequential steps, guiding the model to think through the problem methodically. In the context of industry categorization, the complex task is decomposed into subtasks such as summarizing the organization description, identifying the primary business activity, verifying the industry definition, and providing reasoning for whether the organization matches or does not match the industry. Examples of the chain-of-thought process are also provided in prompting to guide the model. In our trials, we found that including negative examples (illustrating why an organization does not match a particular industry) significantly improves performance compared to offering only positive examples.

To address the potential uncertainty and inconsistency of LLM predictions, we introduce a dual confirmation mechanism. Initially, the LLMs are prompted to select from all categories in the GICSmod and propose potential categories to which the organization may belong. Subsequently, the LLMs are asked multiple times to verify whether the organization indeed fits into the potential category it has suggested. If a potential category is confirmed at least once during the second stage, it will be retained in the final result.

In our trials, relying on a single LLM may yield unsatisfactory outcomes, while combining the results from multiple LLMs leads to more reliable results. A detailed comparison between individual LLMs and combined LLMs can be found in Section 4.2. As a result, the outcomes generated by all LLMs are encoded as vectors and then combined to produce the final industry label. Due to performance variations among the selected LLMs across different categories, distinct weights are assigned to each model's output during the combination process. This combination is performed by a two-layer fully connected neural network, with the weights automatically determined through backpropagation during training on a small, manually annotated dataset.

## 4 EVALUATION OF CATEGORIZATION FRAMEWORK

### 4.1 Gold Standard

We manually constructed a *Gold Standard* dataset as the ground truth for training and evaluating the system. To build a comprehensive dataset, especially when a specific industry is underrepresented, providing candidate categories is essential. Initially, we labeled a sample of 5,000 organizations using five independent LLMs, with the union outputs from these models forming the pool of candidate categories. To ensure a comprehensive representation of all categories, we proportionally sampled one-third of the data from each of these candidates to construct a dataset of 1,500 entries.

Six volunteers were enlisted to label the sampled dataset, with each assigned 500 entries, ensuring that each organization was labeled independently by two volunteers. While the disjoint definitions of GICSmod minimize the chances of a business practice falling into multiple categories, large organizations operating across diverse sectors may still span multiple categories. In such cases, all relevant categories are treated as true labels for the organization, ensuring an accurate reflection of their multi-sector presence.

To minimize the volunteers' workload, they were tasked with verifying the accuracy of the candidate categories rather than selecting from the entire GICSmod. Since most candidate labels consisted of no more than five categories, this approach significantly reduced the effort required compared to manually labeling all 23 categories. In cases of conflicting labels—where one volunteer judged a label as true and another as false—volunteers discussed the disagreement in pairs, with a third volunteer providing adjudication if necessary.

Although this method may overlook certain industries, merging the results from five independent LLMs rarely fails to capture likely industries. The identification of CDN and cloud vendors in Section 3.2 provides insight into these oversights. 97 out of 98 organizations are identified by at least one LLM. The missing sample is Microsoft, which is due to the brief mention of "Microsoft Azure" in the description. Missed categories typically correspond to business aspects that are not prominently mentioned in the description.

Ultimately, we successfully compiled an annotated dataset of 1,328 entries after removing ambiguous descriptions that were insufficient for volunteers to accurately determine the industry. 188 organizations span multiple categories. Each category contains a minimum of 20 organization entries, with the Education category having the highest number at 186.

### 4.2 Evaluation of LLMs and LLMIC

**Settings:** We select distinguished open-source (Llama-2 7B [48], ChatGLM-3 6B [17, 55], Mistral 7B [26]) and close-source (ChatGPT 3.5 [8, 36], Gemini 1.0 [21, 47]) LLMs, publicly available prior to 2024-04, to integrate into the proposed framework. The models are evaluated on the manually labeled gold standard dataset in Section 4.1. For the evaluation of LLMIC, we employ the approach of leave-20%-out cross-validation on the gold standard dataset.

**Table 1: Precision of LLMs and LLMIC on Gold Standard**

| Category | ChatGPT | Gemini | ChatGLM | Llama | Mistral | LLMIC* |
|---|---|---|---|---|---|---|
| Fossil Energy | 15/15(1.00) | 15/18(0.83) | 4/5(0.80) | 4/6(0.67) | 10/11(0.91) | 16/16(1.00) |
| Medical | 58/60(0.97) | 56/59(0.95) | 18/18(1.00) | 35/36(0.97) | 36/38(0.95) | 60/61(0.98) |
| Finance | 46/47(0.98) | 70/73(0.96) | 41/43(0.95) | 68/71(0.96) | 35/38(0.92) | 80/80(1.00) |
| Services | 21/22(0.95) | 19/20(0.95) | 11/16(0.69) | 0/0(0.00) | 13/15(0.87) | 22/22(1.00) |
| Utilities | 22/22(1.00) | 29/30(0.97) | 11/13(0.85) | 19/30(0.63) | 11/13(0.85) | 26/27(0.96) |
| Internet Infrastructure | 23/23(1.00) | 37/40(0.93) | 2/3(0.67) | 26/44(0.59) | 27/34(0.79) | 42/45(0.93) |
| Technological Hardware | 42/43(0.98) | 58/64(0.91) | 3/3(1.00) | 55/105(0.52) | 35/38(0.92) | 65/68(0.96) |
| Media | 59/60(0.98) | 95/95(1.00) | 17/17(1.00) | 110/161(0.68) | 66/66(1.00) | 120/132(0.91) |
| Transportation | 51/52(0.98) | 62/66(0.94) | 13/15(0.87) | 36/40(0.90) | 33/35(0.94) | 61/63(0.97) |
| Defense Manufacturer | 13/13(1.00) | 17/17(1.00) | 2/2(1.00) | 12/18(0.67) | 13/13(1.00) | 16/16(1.00) |
| Fundamental Materials | 19/20(0.95) | 28/30(0.93) | 3/4(0.75) | 1/1(1.00) | 24/29(0.83) | 31/33(0.94) |
| Software | 65/74(0.88) | 73/83(0.88) | 56/99(0.57) | 85/116(0.73) | 56/62(0.90) | 83/88(0.94) |
| Education | 140/143(0.98) | 170/176(0.97) | 93/97(0.96) | 146/147(0.99) | 100/105(0.95) | 182/188(0.97) |
| Capital Goods | 20/21(0.95) | 31/39(0.79) | 22/29(0.76) | 6/7(0.86) | 24/25(0.96) | 37/39(0.95) |
| ISP | 61/64(0.95) | 96/106(0.91) | 76/92(0.83) | 48/53(0.91) | 59/69(0.86) | 94/101(0.93) |
| Government | 120/132(0.91) | 95/101(0.94) | 40/41(0.98) | 51/58(0.88) | 76/86(0.88) | 131/138(0.95) |
| IT Services | 19/20(0.95) | 24/31(0.77) | 2/2(1.00) | 9/12(0.75) | 23/32(0.72) | 28/30(0.93) |
| Automotive | 27/28(0.96) | 29/30(0.97) | 0/1(0.00) | 25/25(1.00) | 21/21(1.00) | 27/28(0.96) |
| Real Estate | 23/24(0.96) | 23/24(0.96) | 9/9(1.00) | 24/42(0.57) | 13/17(0.76) | 26/26(1.00) |
| Retail & Distribution | 28/33(0.85) | 36/43(0.84) | 22/27(0.81) | 29/36(0.81) | 23/40(0.57) | 35/36(0.97) |
| Hotels & Food & Leisure | 19/20(0.95) | 25/29(0.86) | 10/12(0.83) | 3/3(1.00) | 22/23(0.96) | 24/24(1.00) |
| Research | 38/52(0.73) | 41/50(0.82) | 11/11(1.00) | 28/32(0.88) | 28/37(0.76) | 45/49(0.92) |
| Cybersecurity | 25/25(1.00) | 17/17(1.00) | 4/4(1.00) | 10/11(0.91) | 18/20(0.90) | 21/21(1.00) |
| **overall** | **954/1013(0.94)** | **1146/1241(0.92)** | **470/563(0.83)** | **830/1054(0.79)** | **766/867(0.88)** | **1272/1331(0.96)** |

*cross-validated with 20% data left out

**Table 2: The number and proportion of correct samples of LLMs and LLMIC**

| Model | Perfectly labeled samples | Samples with at most one missing label | Samples with at most one incorrect label | Total number of samples with at least one label |
|---|---|---|---|---|
| ChatGPT | 773 (0.83) | 864 (0.93) | 897 (0.96) | 931 |
| Gemini | 930 (0.81) | 1054 (0.92) | 1086 (0.95) | 1146 |
| ChaGLM | 388 (0.71) | 456 (0.83) | 460 (0.84) | 549 |
| Llama | 617 (0.67) | 710 (0.77) | 794 (0.86) | 925 |
| Mistral | 587 (0.76) | 671 (0.87) | 718 (0.93) | 773 |
| LLMIC | 1051 (0.85) | 1166 (0.94) | 1198 (0.97) | 1235 |

The precision results are presented in Table 1. In calculating precision, the task is treated as 23 independent binary classifications, with the precision rate defined as the proportion of correct results among the retrieved predictions. Among the evaluated LLMs, ChatGPT exhibits the highest overall precision rate of 94%, followed by Gemini, Mistral, ChatGLM, and Llama. However, an observation is that even with a high precision rate, there remains a significant amount of unlabeled data points. As for the number of accurate results, Gemini ranks first with 1,146 accurate results, followed by ChatGPT, Llama, Mistral, and ChatGLM. Moreover, the table indicates that each LLM delivers varied performance in various categories, and their ability can complement each other. For example, the performance of ChatGPT in the Research category is suboptimal with only 73% precision, while other models exhibit better results in this category. The observation provides insight into how a combination can capitalize on the unique strengths of all LLMs and augment the overall performance.

By leveraging the capabilities of LLMs in categorizing various industries, LLMIC achieves an overall precision of nearly 96%, accurately classifying 1,272 out of 1,331 samples, surpassing the performance of any individual LLM. The precision across all categories exceeds 90%, with most categories approaching 100%. The system's recall rate stands at 83%, also outperforming individual LLMs. Detailed recall rate results are provided in D.

Since this is a multi-label classification task, we also evaluate the number of correctly labeled samples and their proportion, as summarized in Table 2. The total number of samples refers to those where the model has assigned at least one label, as samples without assigned labels are excluded from further analysis. As shown, LLMIC achieves the best performance by integrating multiple LLMs. 97% of labeled samples have at most one extra or missing label, 94% are fundamentally correct with at most one missing label, and 85% exactly match the ground truth labels. *The above results demonstrate the system's ability to accurately capture diverse industry categories, confirming its reliability for the subsequent analysis.*

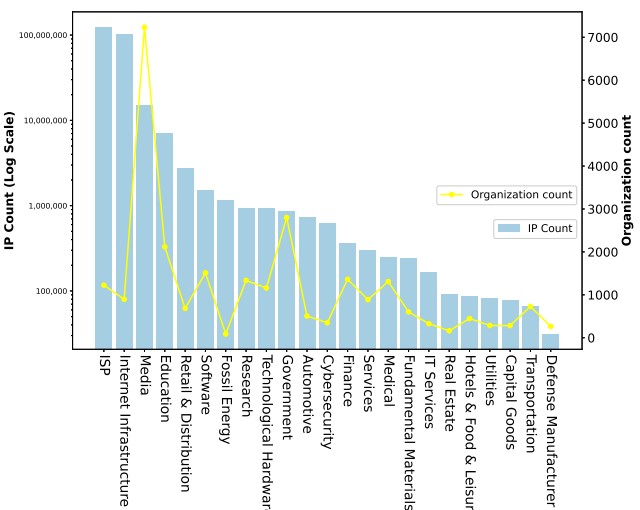

**Figure 3: Industry distribution of labeled IP addresses and organizations. A log scale is applied to the left y-axis.**

**Table 3: The size and distribution of the dataset**

| Source | #IPs | #ASes | #BGP Prefixes |
|--------|------|-------|---------------|
| IPv4 | 253,218,506 | 12,318 | 90,528 |
| IPv6 | 74,6210 | 1,698 | 3,723 |

Moreover, the findings validate the effectiveness of LLMs in expanding datasets from a limited number of annotated samples and offer a flexible framework for leveraging LLMs' strengths in categorizing various categories. The choice of LLMs can be tailored to specific needs. In our experiment, using only closed-source LLMs (i.e., ChatGPT combined with Gemini) resulted in a 5% decrease in correctly labeled entries and a 1% drop in overall precision. By comparison, using only open-source LLMs (i.e., ChatGLM combined with Llama and Mistral) led to a 13% decrease in correctly labeled entries and a 5% drop in overall precision. Thus, while closed-source LLMs offer greater efficiency in terms of the number of models and inference time, open-source LLMs remain a viable option when local deployment is necessary. Detailed results of the ablation study are provided in Appendix C.

## 5 FOOTPRINTS OF WEB SERVICES

### 5.1 Distribution of the Dataset

With the assistance of LLMIC, we have acquired industry categorization for a total of over 200 million IP addresses, encompassing both IPv4 and IPv6. As illustrated in Table 3, these IPv4 addresses are under 12,318 ASes, and 90,528 BGP Prefixes, while IPv6 addresses are under 1,698 ASes and 3,723 BGP Prefixes.

The industry distribution is illustrated in Figure 3, showing that all industries are represented, with the ISP and Internet Infrastructure sectors controlling the majority of identified IP addresses hosting web services. A total of 25,852 organizations were labeled, with the Media industry accounting for the largest number of 7,234 organizations, and the Fossil Energy industry having the fewest of 89. While this method labels organizations with Wikipedia entries,

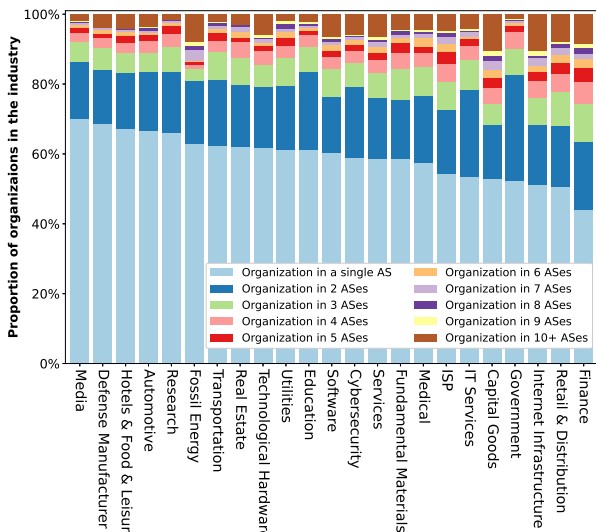

**Figure 4: The proportion of organizations that host web services within a single AS/ multiple ASes.**

it remains representative of the broader industry landscape, as these organizations encompass well-known players within each sector.

### 5.2 Collection of AS information

To further analyze the AS distribution of web services, the data from RouteViews [34] is utilized to discern the AS affiliated with each IP address. To investigate the owner entity of these ASes, we refer to the practices used in ASdb [57] to identify the owner organization. Specifically, we first query RIR records using AS numbers to obtain the associated email addresses and domain names into a domain pool. Then, we remove common email domains (e.g., outlook.com) from the domain pool and eliminate domain names that appear in more than 100 ASes. From the remaining domain names, we identify the organization via the same approach mentioned in 3.2. Finally, we select the *most similar organization* that is the one with the highest similarity to the AS name using cosine similarity between the embeddings encoded by all-MiniLM-L6-v2 [33, 42, 52]. The *most similar organization* is considered as the owner organization of the AS. After the organization is identified, we use the same approach in 3.3 and 3.4 to get the organization information and to determine the industry of the AS owners.

### 5.3 AS Distribution of Web Services

Gigis Petros et al. [20] revealed that Hypergiant companies may deploy services outside their own ASes to minimize the costs associated with crossing network boundaries. Building on this, we further analyze the prevalence of this strategy among organizations across different industries. As illustrated in Figure 4, a significant proportion of organizations across various industries have adopted the strategy of deploying web services across multiple ASes, with the highest rates observed in Finance where around 56% of organizations deploy web services in more than one AS. The Media industry has the lowest proportion of organizations deploying web services across multiple ASes, with 70% of Media organizations using only one AS for their web services. Additionally, a considerable

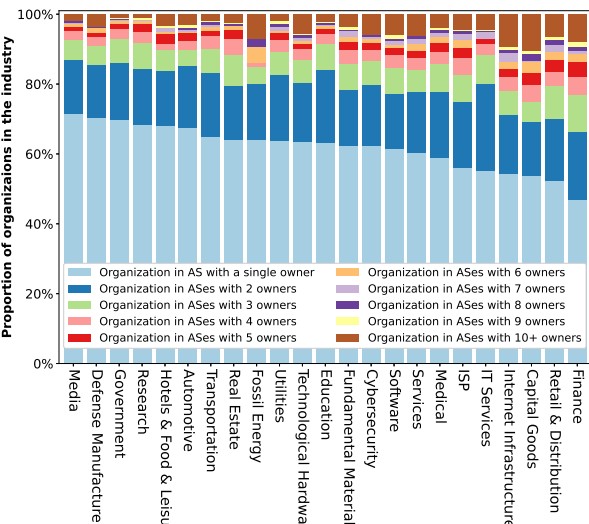

**Figure 5: The proportion of organizations that host web services within AS(es) managed by a single owner/ multiple owners.**

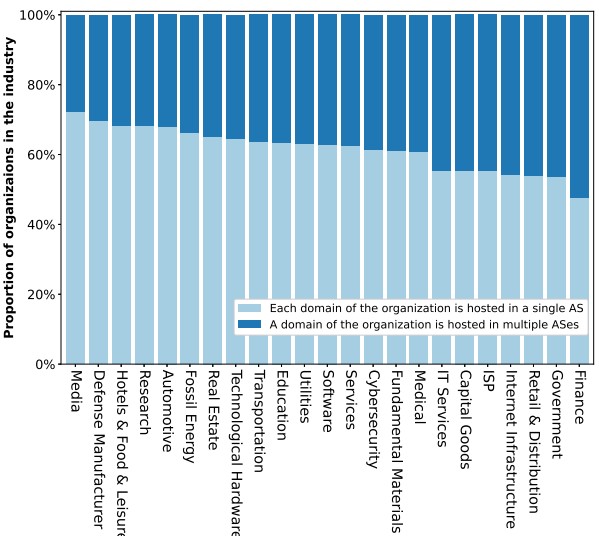

**Figure 6: The proportion of organizations with each domain in a single AS/ one of its domains in multiple ASes.**

number of organizations across various industries have deployed their web services in more than ten ASes.

Considering the owners of these ASes, as illustrated in Figure 5, a significant proportion of organizations host web services within ASes managed by multiple owners. This proportion is comparable to that shown in Figure 4. This observation suggests that organizations do not solely deploy services in multiple ASes owned by a single entity; instead, driven by business needs, they intentionally utilize ASes provided by various providers. Notably, in the ISP and Internet infrastructure sectors, approximately 45% of organizations deploy web services across ASes owned by more than one entity. Given that many organizations in these sectors are also AS owners, this phenomenon corroborates the practice of registered AS organizations renting addresses from other ASes to fulfill user service

requirements. The frequent traffic interactions between addresses within these organizations' own ASes and those external to them offer valuable insights for understanding the Internet ecosystem and modeling cross-AS traffic.

As illustrated in Figure 6, the proportion of organizations with one of its main domains in multiple ASes is also significant across all industries. The slice difference between the proportion in Figure 4 and Figure 6 indicates that the majority of organizations deploying web services across multiple ASes involves not only multiple main domains hosted in different ASes but also a single main domain on multiple IP addresses located in various ASes.

For example, we identified that the well-known automotive company BMW provides web services on 1,944 IP addresses, distributed across 76 ASes managed by 69 AS owners. These IP addresses correspond to 59 main domains, 24 of which have IP addresses spread across multiple ASes. Most domains are registered under the country code top-level domains (ccTLD). According to data from SimilarWeb [43], an established market research platform that provides website traffic analysis, the majority of traffic to these domains originates from the countries or regions to which the ccTLD belongs. Therefore, these domains were registered to serve customers in their respective regions. For ASes hosting these domains, we manually analyzed the countries of the AS owners and found that they generally align with the countries indicated by the domains. For example, bmw.com.cn is registered under China's top-level domain (.cn), and the IP addresses hosting this domain are distributed across six ASes operated by Chinese ISPs. Hosting domains that serve users from a specific country on ASes owned by organizations within that country may aim to enhance service for local users or comply with regulatory requirements. However, considering that the IP addresses for the same domain are distributed across multiple ASes operated by different organizations, it is more likely that it is out of the need to improve user experience.

Combining the data illustrated in Figure 4, 5, and 6, we can infer that the strategy of deploying web services across ASes to reduce cross-border traffic costs and improve user experience is followed by a significant number of organizations across various industries.

### 5.4 Measurement of Cross Industry ASes

As illustrated in Figure 7, when examining the industry of AS owners where critical infrastructure organizations deploy their web services, it becomes clear that, across all industries, web services are sometimes hosted in ASes managed by organizations outside their industry. The Fossil Energy sector exhibits the lowest proportion of web services deployed in ASes owned by other industries, followed by ISP and Internet Infrastructure. Furthermore, when organizations from most industries host their web services in ASes outside their industry, they tend to favor ASes owned by ISPs.

From an AS-level perspective, the results reveal a widespread occurrence of ASes hosting web services from industries different from those of the AS owners. Specifically, 87.83% of IPv4 ASes and 72.96% of IPv6 ASes host web services from industries that are not reflected in the AS owner's industry label, as shown in Figure 8.

For example, AS 28760, owned by Infotech EDV-Systeme GmbH, an IT service provider in Austria, is identified as hosting 15,733 IP addresses for web services belonging to the same organization,

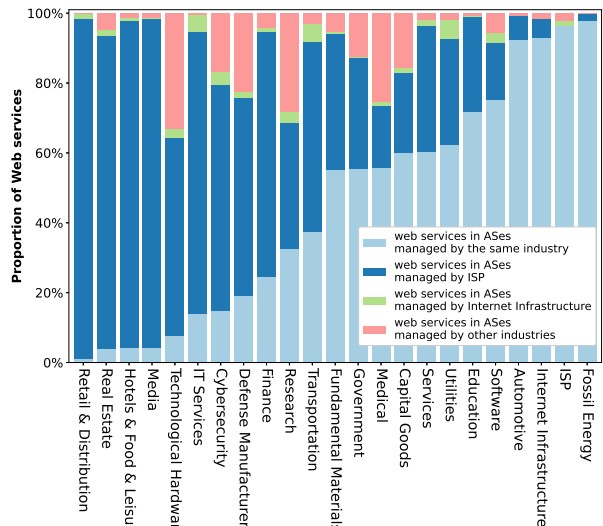

**Figure 7: The proportion of web services (count by FQDN) deployed in ASes owned by the same industry, by ISP, by Internet Infrastructure provider (except ISP), and owners from other industries.**

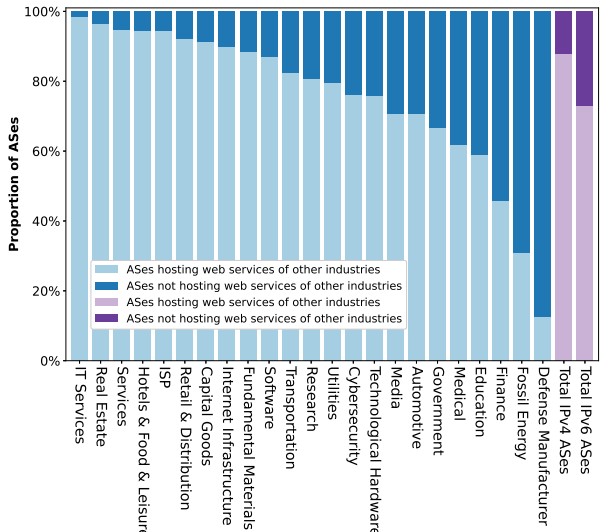

**Figure 8: The proportion of ASes that host web services of other industries**

along with 6 IP addresses hosting services for three other organizations. These include an international automotive manufacturer, an aerospace design company headquartered in Austria, and an Austrian lighting studio. The domains associated with these IP addresses are registered under Austria's country code top-level domain (.at), and for two of these domains, the AS owner is also the registrar, as indicated by domain WHOIS records. According to Similarweb statistics [43], the majority of traffic to these three domains originates from Austria (87%–100%). This example illustrates how hosting other organizations' web services within an AS can provide support for smaller organizations or those lacking their own AS infrastructure, helping them bring web services online, while

also enhancing the quality of service distribution by considering the geographic location of users. Although deploying web services within their own AS fosters a mutually beneficial web ecosystem for both web service providers and AS owners, it also introduces challenges for regulatory compliance and security management, as the true end-user of the IP addresses becomes difficult to identify.

Figure 8 highlights the widespread presence of cross-industry phenomenon across ASes owned by all industries. The Defense Manufacturer sector shows the lowest occurrence of this phenomenon, while ASes owned by IT services exhibit the highest proportion of cross-industry web hosting. Most sectors, except for Finance, Fossil Energy, and Defense manufacturers, show that over 50% of their ASes host web services from industries other than their own. The analysis further reveals that ASes managed by all industries, including those in critical infrastructure, frequently host web services from different sectors. Consequently, relying solely on the AS owner to infer the ownership of IP addresses, particularly when identifying entities behind risky or malicious addresses, can lead to significant inaccuracies in IP address security research.

## 6 RELATED WORK

**AS to Industry.** Dimitropoulos et al. [16] categorize AS owners into six groups, such as Large ISPs, Small ISPs, and Universities, using organization description and topology. Dhamdhere et al. [15] classify ASes based on their topology. Baumann et al. [5] employ keyword analysis of WHOIS data to sort ASes into 10 categories. Ziv et al. [57] introduce NAICSlite with 95 categories, utilizing business intelligence databases and machine learning for classification.

**Website to Industry.** There are existing efforts in Artificial Intelligence to classify websites into industries. For instance, López-Sánchez et al. [30] categorize web pages by visual content. Bruni et al. [9] utilize machine learning approaches to identify e-commerce websites. As they do not focus on Internet management, the key problem lies in the acquisition of data for adapting their methods.

**Organization to Industry.** Established business intelligence databases [7, 12, 13] have been evaluated for network classification [57]. However, these databases primarily focus on commercial aspects rather than network management, leaving a noticeable gap for application. Moreover, their relatively low accuracy in categorizing technological entities [57] presents further challenges.

## 7 CONCLUSION

In this paper, we propose an LLM-based industry categorization system with a precision rate of nearly 96%. Using this system, we built an IP-level industry categorization dataset, IPdb, which encompasses 200 million IP addresses from 12,318 IPv4 ASes and 1,698 IPv6 ASes. Through IPdb, we analyze how organizations across various industries strategically deploy web services across multiple ASes to enhance service delivery. Our findings also reveal cross-industry phenomena within ASes, underscoring the value of IP-level categorization in gaining deeper insights into the Internet ecosystem. While IPdb's organization-based approach has some limitations in fully categorizing all web service-related IP addresses, it provides crucial annotated data for more granular categorization across the network.

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

## A ETHICS

For active scanning to get certificates and PTR Records, and scraping to get information from Wikipedia, we took care to scan at a low rate to minimize the potential harm to the routers, networks, and destination websites. Following the best current practices [14, 18, 37], we configure our prober to run a website on 80 port, with experiment information. It provides a contact e-mail address to request exclusion from scans. We did not receive any complaints or requests during our study. For external data, we applied for research accounts to Censys. The accounts allowed us to query and download the data.

## B DESCRIPTION OF GICSMOD

## C ABLATION STUDY OF SELECTED LLMS IN THE FRAMEWORK

## D RECALL OF SELECTED LLMS AND LLMIC

**Table 4: Description of GICSmod**

| Category | Description |
|---|---|
| Medical | Medical care institutions, biotechnology, pharmaceuticals, life sciences and healthcare technology. Excluding drug retail, health insurance, medical magazines |
| Finance | Finance institutions, including banks, insurance, investment, mortgage REITs, and other financial business. Excluding managed care |
| Fossil Energy | Fossil energy companies, including coal, oil & gas drilling, exploration, production and refining |
| Utilities | Companies that produce or distribute electricity, natural, manufactured gas, and water to end consumers. exclude petrochemicals. |
| Internet Infrastructure | Internet infrastructure & services, including data centers, storage infrastructure, web hosting, cloud servers, content delivery network, domain name registrar, public key infrastructure. |
| Media | Company involved in content creation and distribution including advertising, publishing, producers of movies, television, music, or games or broadcast programs, as well as online platforms like social media. |
| Transportation | Transportation companies and transportation infrastructure operators |
| Defense Manufacturer | Defense industry companies.exclude military, government agencies, security companies. |
| Education | Schools, colleges, universities, and other educational organizations or services |
| Research | Research and development organizations |
| ISP | Internet service providers |
| Government | Government and regulatory agencies, administrations, departments, and military |
| Real Estate | Real estate operating, development & services, and real estate investment trusts (REITs), excluding mortgage REITs |
| Retail & Distribution | Retailers, distributors, e-commerce, and internet retail |
| Services | Services including human resources and employment, consulting services, information vendor, administrative, support, waste management and remediation services, legal, accounting, tax preparation, bookkeeping, payroll, design services, commercial printing, management, etc |
| Technological Hardware | Manufacturers of communication equipment (routers, switches), storage (hard drives, memory), and peripheral devices (mice, keyboards), mobile phones, computers, instruments, semiconductors, photovoltaic materials, also including electronic product manufacturing services |
| Fundamental Materials | Materials and chemical companies, including chemicals, industrial gases, construction materials, building products, forest products, home furnishings, metal, glass & plastic containers, paper & plastic packaging products & materials, paper products, housewares & specialties, metals & mining |
| Software | Software developers. Exclude games (Media), data centers, storage infrastructure, web hosting, cloud servers, content delivery network, domain name registrar, public key infrastructure (Internet Infrastructure), cyber security services (Cybersecurity), data recovery, system integration services, data processing and outsourcing services(IT Services) |
| Capital Goods | manufacturers of machinery, equipment, and components, building products, aerospace, construction, engineering, electrical components, heavy electrical equipment, construction machinery, heavy transportation equipment, agricultural and farm machinery, and industrial machinery. include industrial automation |
| Automotive | Manufacturers of automobiles, motorcycles, and its accessories |
| Hotels & Food & Leisure | Recreation & accommodation & food services, including casinos & gaming, hotels, resorts, restaurants, leisure facilities |
| IT Services | Computer systems design and related services, including computer facilities management services, custom computer programming services, computer systems integration design services, computer hardware or software consulting services, software installation services |
| Cybersecurity | computer, network and cyberspace security services and software companies |
| Other | Other industries not listed above |

**Table 5: Ablation Study of integrated LLMs in the LLMIC Framework**

| Category | non-ChatGPT | non-Gemini | non-ChatGLM | non-Llama | non-Mistral | Open-source | Close-source |
|---|---|---|---|---|---|---|---|
| Fossil Energy | 12/13(0.92) | 14/14(1.00) | 17/17(1.00) | 15/16(0.94) | 14/14(1.00) | 9/9(1.00) | 13/14(0.93) |
| Medical | 57/57(1.00) | 59/62(0.95) | 61/63(0.97) | 62/64(0.97) | 62/62(1.00) | 45/47(0.96) | 61/62(0.98) |
| Finance | 80/83(0.96) | 75/78(0.96) | 80/82(0.98) | 76/79(0.96) | 79/81(0.98) | 75/78(0.96) | 72/76(0.95) |
| Services | 19/20(0.95) | 22/23(0.96) | 21/21(1.00) | 24/25(0.96) | 25/28(0.89) | 15/17(0.88) | 25/27(0.93) |
| Utilities | 26/27(0.96) | 26/27(0.96) | 27/28(0.96) | 25/25(1.00) | 26/26(1.00) | 22/26(0.85) | 27/27(1.00) |
| Internet Infrastructure | 41/47(0.87) | 40/44(0.91) | 44/47(0.94) | 43/46(0.93) | 43/45(0.96) | 34/41(0.83) | 43/45(0.96) |
| Technological Hardware | 59/63(0.94) | 66/67(0.99) | 63/66(0.95) | 64/67(0.96) | 64/69(0.93) | 56/68(0.82) | 61/67(0.91) |
| Media | 118/129(0.91) | 115/125(0.92) | 120/127(0.94) | 106/106(1.00) | 115/127(0.91) | 116/132(0.88) | 101/102(0.99) |
| Transportation | 60/63(0.95) | 56/59(0.95) | 62/66(0.94) | 62/65(0.95) | 59/64(0.92) | 49/54(0.91) | 59/61(0.97) |
| Defense Manufacturer | 15/15(1.00) | 15/15(1.00) | 16/16(1.00) | 16/16(1.00) | 16/16(1.00) | 13/13(1.00) | 14/14(1.00) |
| Fundamental Materials | 28/32(0.88) | 28/30(0.93) | 32/35(0.91) | 31/34(0.91) | 27/29(0.93) | 21/26(0.81) | 30/32(0.94) |
| Software | 79/86(0.92) | 82/87(0.94) | 80/84(0.95) | 77/85(0.91) | 80/90(0.89) | 81/92(0.88) | 76/83(0.92) |
| Education | 181/187(0.97) | 177/182(0.97) | 180/184(0.98) | 179/188(0.95) | 183/188(0.97) | 177/182(0.97) | 181/188(0.96) |
| Capital Goods | 35/39(0.90) | 35/40(0.88) | 35/39(0.90) | 36/38(0.95) | 32/40(0.80) | 32/37(0.86) | 31/35(0.89) |
| ISP | 96/102(0.94) | 86/92(0.93) | 90/97(0.93) | 96/102(0.94) | 96/102(0.94) | 89/101(0.88) | 92/100(0.92) |
| Government | 118/128(0.92) | 123/131(0.94) | 132/141(0.94) | 130/137(0.95) | 126/130(0.97) | 102/110(0.93) | 127/132(0.96) |
| IT Services | 21/23(0.91) | 23/28(0.82) | 25/30(0.83) | 26/29(0.90) | 25/28(0.89) | 17/19(0.89) | 23/27(0.85) |
| Automotive | 30/30(1.00) | 29/29(1.00) | 29/29(1.00) | 29/29(1.00) | 29/29(1.00) | 29/29(1.00) | 30/31(0.97) |
| Real Estate | 22/22(1.00) | 24/25(0.96) | 24/25(0.96) | 24/24(1.00) | 26/26(1.00) | 16/17(0.94) | 25/26(0.96) |
| Retail & Distribution | 34/35(0.97) | 30/31(0.97) | 31/35(0.89) | 32/37(0.86) | 32/33(0.97) | 30/31(0.97) | 32/37(0.86) |
| Hotels & Food & Leisure | 24/24(1.00) | 25/26(0.96) | 25/25(1.00) | 25/25(1.00) | 25/26(0.96) | 23/23(1.00) | 26/26(1.00) |
| Research | 42/46(0.91) | 42/46(0.91) | 45/52(0.87) | 47/51(0.92) | 40/44(0.91) | 35/38(0.92) | 37/40(0.93) |
| Cybersecurity | 18/21(0.86) | 21/22(0.95) | 26/26(1.00) | 24/24(1.00) | 23/23(1.00) | 16/18(0.89) | 23/23(1.00) |
| **overall** | **1215/1292(0.94)** | **1213/1283(0.95)** | **1265/1335(0.95)** | **1249/1312(0.95)** | **1247/1320(0.94)** | **1102/1208(0.91)** | **1209/1275(0.95)** |

**Table 6: Recall of LLMs and LLMIC on Gold Standard**

| Category | ChatGPT | Gemini | ChatGLM | Llama | Mistral | LLMIC* |
|---|---|---|---|---|---|---|
| Fossil Energy | 15/20(0.75) | 15/20(0.75) | 4/20(0.20) | 4/20(0.20) | 10/20(0.50) | 16/20(0.80) |
| Medical | 58/79(0.73) | 56/79(0.71) | 18/79(0.23) | 35/79(0.44) | 36/79(0.46) | 60/79(0.76) |
| Finance | 46/90(0.51) | 70/90(0.78) | 41/90(0.46) | 68/90(0.76) | 35/90(0.39) | 80/90(0.89) |
| Services | 21/33(0.64) | 19/33(0.58) | 11/33(0.33) | 0/33(0.00) | 13/33(0.39) | 22/33(0.67) |
| Utilities | 22/34(0.65) | 29/34(0.85) | 11/34(0.32) | 19/34(0.56) | 11/34(0.32) | 26/34(0.76) |
| Internet Infrastructure | 23/61(0.38) | 37/61(0.61) | 2/61(0.03) | 26/61(0.43) | 27/61(0.44) | 42/61(0.69) |
| Technological Hardware | 42/82(0.51) | 58/82(0.71) | 3/82(0.04) | 55/82(0.67) | 35/82(0.43) | 65/82(0.79) |
| Media | 59/139(0.42) | 95/139(0.68) | 17/139(0.12) | 110/139(0.79) | 66/139(0.47) | 120/139(0.86) |
| Transportation | 51/77(0.66) | 62/77(0.81) | 13/77(0.17) | 36/77(0.47) | 33/77(0.43) | 61/77(0.79) |
| Defense Manufacturer | 13/22(0.59) | 17/22(0.77) | 2/22(0.09) | 12/22(0.55) | 13/22(0.59) | 16/22(0.73) |
| Fundamental Materials | 19/42(0.45) | 28/42(0.67) | 3/42(0.07) | 1/42(0.02) | 24/42(0.57) | 31/42(0.74) |
| Software | 65/102(0.64) | 73/102(0.72) | 56/102(0.55) | 85/102(0.83) | 56/102(0.55) | 83/102(0.81) |
| Education | 140/186(0.75) | 170/186(0.91) | 93/186(0.50) | 146/186(0.78) | 100/186(0.54) | 182/186(0.98) |
| Capital Goods | 20/58(0.34) | 31/58(0.53) | 22/58(0.38) | 6/58(0.10) | 24/58(0.41) | 37/58(0.64) |
| ISP | 61/110(0.55) | 96/110(0.87) | 76/110(0.69) | 48/110(0.44) | 59/110(0.54) | 94/110(0.85) |
| Government | 120/148(0.81) | 95/148(0.64) | 40/148(0.27) | 51/148(0.34) | 76/148(0.51) | 131/148(0.89) |
| IT Services | 19/37(0.51) | 24/37(0.65) | 2/37(0.05) | 9/37(0.24) | 23/37(0.62) | 28/37(0.76) |
| Automotive | 27/37(0.73) | 29/37(0.78) | 0/37(0.00) | 25/37(0.68) | 21/37(0.57) | 27/37(0.73) |
| Real Estate | 23/27(0.85) | 23/27(0.85) | 9/27(0.33) | 24/27(0.89) | 13/27(0.48) | 26/27(0.96) |
| Retail & Distribution | 28/38(0.74) | 36/38(0.95) | 22/38(0.58) | 29/38(0.76) | 23/38(0.61) | 35/38(0.92) |
| Hotels & Food & Leisure | 19/28(0.68) | 25/28(0.89) | 10/28(0.36) | 3/28(0.11) | 22/28(0.79) | 24/28(0.86) |
| Research | 38/58(0.66) | 41/58(0.71) | 11/58(0.19) | 28/58(0.48) | 28/58(0.48) | 45/58(0.78) |
| Cybersecurity | 25/28(0.89) | 17/28(0.61) | 4/28(0.14) | 10/28(0.36) | 18/28(0.64) | 21/28(0.75) |
| **overall** | **954/1536(0.62)** | **1146/1536(0.75)** | **470/1536(0.31)** | **830/1536(0.54)** | **766/1536(0.50)** | **1272/1536(0.83)** |

*cross-validated with 20% data left out

