# OpenReview forum: "IPdb: A High-precision IP Level Industry Categorization of Web Services"
_ACM.org/TheWebConf/2025/Conference — WWW 2025 Poster_

### Official Review · Reviewer_8RvU · 2024-11-15

**Novelty:** 3
**Technical Quality:** 4

**Review:**

This paper proposes an LLM-based industry categorization system for IP-level industry categorization of IP addresses. Using the proposed model, an IP-level industry categorization dataset, IPdb, is built. Experimental studies draw some inspiring findings by analyzing the web service deployment strategy of organizations across various industries. We paper is well written and the method is clearly introduced.

The model mainly contains four parts, domain identification, organization identification, web description collection, and organization classification. However, innovations of the method may need rethinking.

**Questions:**

In domain identification, the authors utilize DNS PTR records to identify domains. In organization identification, they perform domain WHOIS lookups. While in web description collection, they scrape text info from Wikipedia. Last, In the organization classification procedure, ensembled LLM models with dynamical ensembling weights are built to make classification based on text descriptions. However, all the above are a combination of existing technologies, carried out sequentially to fulfil the IP-level industry categorization task. What are the  innovations in the above procedures?

**Reviewer Confidence:**

3: The reviewer is confident but not certain that the evaluation is correct

**Scope:**

4: The work is relevant to the Web and to the track, and is of broad interest to the community

---

### Official Review · Reviewer_x2XX · 2024-11-18

**Novelty:** 5
**Technical Quality:** 5

**Review:**

Summary: this work leverages the powerful capacities of multiple large language models to create a more granular IP-level industry categorization framework than traditional AS-level ones. With this framework, they generated a dataset with 200 million IP addresses.

Pros:

1 - Based on the claimed contributions, the paper appears to make meaningful and satisfactory contributions to the relevant fields.

2- The experimental results presented in the paper are comprehensive and highly impressive. They effectively demonstrate the robustness and applicability of the proposed methods.

Cons:

1 - Some abbreviations, such as "BGP" on page 1, are introduced without explanation, which might confuse readers unfamiliar with the terms. It would be helpful to provide definitions or explanations for these abbreviations when they first appear.

2 - Section 3 merely describes the methodology for categorization, which does not provide sufficient detail for readers to fully understand Figure 1. However, some details about Figure 1 are only provided at the end of Section 3.

3 - The placement of the related work section immediately before the conclusion feels unconventional and disrupts the logical flow of the paper. Furthermore, the related work is presented too concisely, lacking detailed comparisons with existing techniques or sufficient discussion of recent advancements. This brevity makes it unclear whether the paper adequately covers the state-of-the-art research in the field. Additionally, some of the referenced work is over ten years old, raising concerns about the relevance and timeliness of the cited studies. A more thorough and up-to-date review of related work, placed earlier in the paper, would help establish the context, highlight the gaps in existing research, and better support the novelty of the proposed contributions.

**Questions:**

I am not an expert in this research field. My only concern is that the related work is presented too concisely, lacking detailed comparisons with existing techniques or sufficient discussion of recent advancements. This brevity makes it unclear whether the paper adequately covers the state-of-the-art research in the field.

**Reviewer Confidence:**

3: The reviewer is confident but not certain that the evaluation is correct

**Scope:**

4: The work is relevant to the Web and to the track, and is of broad interest to the community

---

### Official Review · Reviewer_Aw7y · 2024-11-20

**Novelty:** 5
**Technical Quality:** 6

**Review:**

This paper introduces a large dataset of IP-addresses categorized by industries. The automated methodology proposed involves an initial reverse DNS lookup to obtain the organization identification, which is then used to scrap the organization information from wikipedia. The final step involves prompting the chosen LLM to categorize the organization. The authors conducted human evaluations on a meaningful number of organizations across five different LLMs, evaluating their automated process to be high in precision. The authors also included discussions on choice of LLMs, and an analysis of the data mined. In doing so, the authors discover a degree of misclassification at the autonomous systems. Other possible downstream implications may involve cost-optimizations for organizations.

Pros:
+ Contributes IPdb: an IP-level industry categorization dataset comprising 200 million IPv4 & IPv6 addresses
+ Proposes LLM-augmented automated classification process, evaluated to be high precision
+ "Gold Standard" evaluation using human evaluators
+ Discussion on the implications of using LLMs may interest a wider audience

Overall, the paper is very clear and well-written. My opinion is that the paper is interesting and executed well, albeit using a known strategy of LLM for classification.

**Questions:**

A) From my understanding, organizations may change their IP addresses, so is this dataset meant to be static or continually updated? If it is not updated, how does the dataset remain relevant into the future?

B) One potential use case is for cybersecurity research; if it is continuously updated, are there any implications if the dataset is poisoned by bad actors exploiting the hallucinations of LLMs?

**Reviewer Confidence:**

3: The reviewer is confident but not certain that the evaluation is correct

**Scope:**

4: The work is relevant to the Web and to the track, and is of broad interest to the community

---

### Official Review · Reviewer_gTbv · 2024-11-25

**Novelty:** 4
**Technical Quality:** 4

**Review:**

This paper aims to address the lack of annotated IP-level datasets. To address this challenge, this paper presents IPdb, an IP-level industry categorization dataset. To construct the dataset, this paper develops LLMIC, a Large Language Model-based Industry Categorization framework with a high precision. The authors also analyze how organizations across various industries strategically deploy web services across multiple ASes to enhance service delivery.

Strengths.

[+] The writing of this paper is easy to implement.

[+] This paper studies a practical challenge with potential applications in many domains.

[+] This paper provides the discrepancy analysis between IP-level and AS-level industry categorization.

Weaknesses

[-] In the proposed method, the authors design a large language model-based framework. However, the authors fail to consider the uncertainties in the outputs generated by large language models. Considering uncertainty is critical because when the model is highly confident about its outputs, it suggests that the information provided is likely more reliable and informative. Conversely, uncertain outputs may require additional scrutiny or complementary methods to ensure robustness and accuracy.

[-] It is unclear how to effectively check and evaluate the correctness of the outputs generated by large language models. Existing literature highlights that LLMs are highly vulnerable to small perturbations, which can lead to significant changes in their outputs. This vulnerability suggests that the models can be unstable and unreliable. The stability is also an very important factor for the proposed framework. Additionally, more discussions on how to incorporate this factor to adjust informative importance scores for the adopted multiple large language models should be given.

[-] Does traditional data augmentation methods and synthetic data generation methods (e.g., GAN based methods and diffusion models based methods) could improve the annotation process?

[-] For the annotated labeled dataset, how to quantify the uncertainty behind each label annotation? Quantifying and addressing this label uncertainty is also of critical importance. Additionally, it would be valuable to evaluate other important aspects of model performance, such as fairness, robustness, and each feature’s informativeness.

**Questions:**

See the weaknesses in the above.

**Reviewer Confidence:**

2: The reviewer is willing to defend the evaluation, but it is likely that the reviewer did not understand parts of the paper

**Scope:**

3: The work is somewhat relevant to the Web and to the track, and is of narrow interest to a sub-community

---

### Official Review · Reviewer_mVfp · 2024-12-02

**Novelty:** 4
**Technical Quality:** 4

**Review:**

The paper introduces IPdb, a high-precision dataset that categorizes over 200 million IP addresses into industries using the LLMIC framework, which combines multiple Large Language Models (LLMs) and external techniques like reverse DNS, WHOIS, and SSL certificate analysis.

+ The paper is overall quite clear and organized very well to explain all the parts of the proposed LLMIC framework.
+ The work addresses a gap of IP-level datasets in the literature that helps to analyze how organizations in different industries deploy their servises across multiple ASes or how they host other organization’s web services in their ASes.
+ The motivation behind this work is also very well explained in the paper by focusing on more cyberattack concerns and internet management. However, to strength the motivation, it would be also good if the authors share insights about more use cases like how it can be useful for network optimization and traffic analysis operations like bandwidth prioritization and load balancing.

-	The gold standard dataset, used for training and evaluation of the system, with 1,328 entries is relatively small compared to the scale of the IPdb dataset with 200 million IP addresses. It may create a problem on the proposed framework’s generalizability since it may not fully capture the diversity of real-world scenarios.
-	Despite high precision of 96%, the recall rate of 83% indicates that a significant number of relevant classifications are missed, that may affect negatively the applications requiring comprehensive industry coverage.

**Questions:**

1)	To better evaluate the framework’s generalizability, the authors can consider expanding the gold standard dataset.
2)	In table 1, why the total number of samples evaluated is different over different LLMs and LLMIC?
3)	What are the most common types of misclassifications in the gold standard dataset? Are these errors caused by the limitations in the LLMs, insufficient training data, or inaccuracies in techniques like reverse DNS, WHOIS, and SSL certificate analysis? It would be good if the authors share more insights about this in the paper.
4)	The paper discusses the widespread use of ASes across multiple industries. Could you share some insights about how this affects cybersecurity and regulatory compliance?

**Reviewer Confidence:**

3: The reviewer is confident but not certain that the evaluation is correct

**Scope:**

3: The work is somewhat relevant to the Web and to the track, and is of narrow interest to a sub-community